# ReFACT: Updating Text-to-Image Models by Editing the Text Encoder

## ABSTRACT

Our world is marked by unprecedented technological, global, and socio-political transformations, posing a significant challenge to text-to-image generative models. These models encode factual associations within their parameters that can quickly become outdated, diminishing their utility for end-users. To that end, we introduce ReFACT, a novel approach for editing factual associations in text-to-image models without relaying on explicit input from end-users or costly re-training. ReFACT updates the weights of a specific layer in the text encoder, modifying only a tiny portion of the model's parameters and leaving the rest of the model unaffected. We empirically evaluate ReFACT on an existing benchmark, alongside a newly curated dataset. Compared to other methods, ReFACT achieves superior performance in both generalization to related concepts and preservation of unrelated concepts. Furthermore, ReFACT maintains image generation quality, making it a practical tool for updating and correcting factual information in text-to-image models.[1]

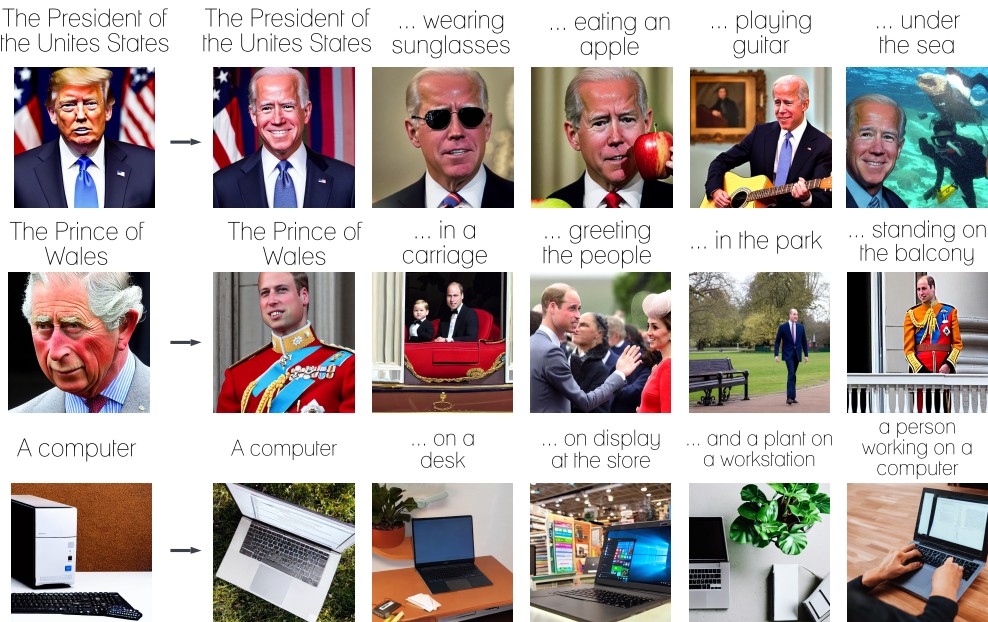

Figure 1: ReFACT edits knowledge in text-to-image models using an editing prompt and a target prompt (e.g., "The President of the United States" is edited to "Joe Biden"), such that the edit is generlizable to other related prompts unseen during editing.

---

[1]Our code and data can be found in the supplementary material, and will be made publicly available.

# 1 INTRODUCTION

Text-to-image generative models (Ho et al., 2022; Dhariwal & Nichol, 2021; Ramesh et al., 2022; Rombach et al., 2022) are trained on extensive amounts of data, leading them to implicitly encode factual associations within their parameters. While some facts are useful, others may be incorrect or become outdated (e.g., the current President of the United States; see Figure 1). Once these models have been trained, they quickly become outdated and misrepresent the state of the world in their generations. However, model providers and creators currently have no efficient means to update them without either retraining them—which is costly in computation and time—or requiring explicit prompt engineering from the end user.

In this work we present ReFACT, a new method for **R**evising **FACT**ual knowledge in text-to-image models. ReFACT views facts as key–value pairs encoded in the linear layers of the transformer and updates the weights of a specific layer in the text-encoder by editing a key–value mapping using a closed form solution (Meng et al., 2022a). Our method utilizes three textual inputs: an edit prompt, a source, and a target, representing the desired edit. For example, "The President of the United States" as the edit prompt, "Donald Trump" as the source, and "Joe Biden" as the target. Then, an edit can be viewed as changing the value the model retrieves for the corresponding key ("The President of the United States") from source to target ("Donald Trump" → "Joe Biden"). By doing so, ReFACT edits the factual associations of the model without fine-tuning. ReFACT modifies only a tiny portion of the model's parameters (0.24%), far fewer than the previous editing method, TIME (1.95%).

Once ReFACT is applied to the model, we achieve a persistent change in factual information, resulting in a model that consistently generates images of Joe Biden for the desired prompt. Moreover, ReFACT is able to generalize to closely related prompts and demonstrate the desire update, while not affecting unrelated concepts. Notably, ReFACT preserves the general quality of generated images.

We evaluate ReFACT on the TIME dataset (Orgad et al., 2023), a benchmark for evaluating the editing of implicit model assumptions on specific attributes (e.g., editing roses to be blue instead of red). Furthermore, we curate a new dataset, the **Ro**les and **A**ppearances **D**ataset (RoAD), for editing other types of factual associations. We show that ReFACT successfully edits a wide range of factual association types, demonstrates high generalization, and does not hurt the representations of unrelated facts. Our method achieves superior results compared to a recently proposed editing method, TIME (Orgad et al., 2023). Overall, our method is a significant improvement in text-to-image model editing.

# 2 RELATED WORK

Editing knowledge embedded within deep neural networks has been the focus of several lines of work, achieving success in editing generative adversarial networks (Bau et al., 2020; Nobari et al., 2021; Wang et al., 2022), image classifiers (Santurkar et al., 2021), and large language models (LLMs) (Meng et al., 2022b; Raunak & Menezes, 2022; Mitchell et al., 2021). Several methods were proposed to update weights in LLMs in particular, including fine-tuning on edited facts (Zhu et al., 2020), weight predictions using hyper-networks (Cao et al., 2021), identifying and editing specific neurons (Dai et al., 2021), and rank one model editing (Meng et al., 2022a). The task of factual editing in text-to-image models was introduced by Orgad et al. (2023), who targeted the cross-attention layers. In contrast, we target a specific layer in the text encoder of the text-to-image model, allowing a more precise edit that changes fewer model parameters (0.24% compared to 1.95% of the model parameters) and outperforms on all metrics.

The task of editing knowledge in (the parameters of) text-to-image models is separate from two other lines of work. First, a large body of work has been devoted to *image editing* (Avrahami et al., 2022; Mokady et al., 2022; Nichol et al., 2021; Wallace et al., 2022; Wu & De la Torre, 2022; Zhang et al., 2022; Couairon et al., 2022). Image editing aims to modify specific attributes of an input image based on some auxiliary inputs, recently using texts and instructions (Bau et al., 2021; Kawar et al., 2022; Hertz et al., 2022). Contrary to our setting, this task does not aim to make a persistent change in the models' generations and thus does not consider modification of the model's weights.

Another distinct line of work is personalization of text-to-image diffusion models, where the goal is to adapt the model to a specific individual or object (Agrawal et al., 2021; Ruiz et al., 2022). Personalization in text-to-image models allow the model to better generate a specific face, object, or scene, given a specific word or pseudo-word (Cohen et al., 2022; Gal et al., 2022; Daras & Dimakis, 2022; Tewel et al., 2023). Personalization methods provide the user with a new token or embedding

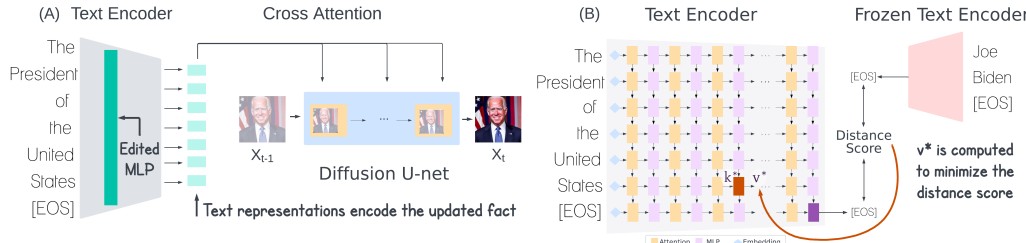

Figure 2: **(A)** An overview of a diffusion text-to-image model after editing with ReFACT. The edited text encoder generates textual representations reflecting the updated information. Then, the representations are fed into the cross-attention mechanism of a diffusion model, generating an image reflecting the new fact. **(B)** ReFACT receives an edit prompt and a target prompt representing the desired change. We obtain the representation of the target and other contrastive examples by passing it through the frozen CLIP text encoder and taking the output at the [EOS] token. Then, we optimize a vector $v^*$ that, when inserted in a specific layer, will reduce the distance between the edit and the target prompts representation, and increase the distance with respect to the contrastive examples. The vector $v^*$ is then planted in the MLP layer using a closed form solution.

that represents a novel entity, while preserving the original class of objects (for example, utilizing the special token "[v]" to represent a specific dog in the prompt "A [v] dog", while preserving the original generic meaning of "A dog"). In contrast, our work focuses on a fundamentally different task: completely transforming the factual associations without preserving the original value. For example, after editing, the model should consistently generate images of Joe Biden for all prompts and phrases related to "The President of the United States", without the need to include a special token in the user's prompt, and without preserving the original outdated association to Donald Trump. Thus, our method provides a practical way for model providers to keep their models up to date.

## 3 METHOD

### 3.1 BACKGROUND

Diffusion models generate images by gradually denoising noisy samples until a clean image is obtained. Text-to-image diffusion models (Rombach et al., 2022; Ramesh et al., 2022; Saharia et al., 2022b) are conditioned on a text prompt that guides the image generation process. Several text-to-image diffusion models use CLIP (Radford et al., 2021) as a multi-modal-aware text encoder.

CLIP consists of a text encoder and an image encoder, jointly trained to create a shared embedding space for images and texts. Concretely, a special end-of-sequence token, denoted [EOS], is appended at the end of each input. CLIP is trained contrastively to maximize the cosine similarity between [EOS] token representations of corresponding texts and images while minimizing the similarity between unrelated inputs. CLIP's text encoder is a transformer model with a GPT-2 style architecture (Radford et al., 2018) trained from scratch. Since the text encoder implements a causal (unidirectional) attention mechanism, the [EOS] is the only token able to aggregate information from all other tokens in the sequence. Thus, the [EOS] token is suitable for optimizing the insertion of new facts.

### 3.2 REFACT

Since the image generation process is conditioned on the representations produced by the text encoder, we hypothesize that editing the knowledge of the text encoder should be reflected in the generated images. At a high level, ReFACT takes an edit prompt (e.g., "The President of the United States"), and source and target prompts that reflects the desired edit ("Donald Trump" → "Joe Biden"), and edits a specific layer in the text encoder. The goal is to make the model's representation of the edit prompt similar to that of the target prompt, in contrast to the representation of the source prompt. The process is illustrated in Figure 2.

To achieve this, ReFACT targets the multi-layer perceptron (MLP) layers in the text encoder. Each MLP consists of two matrices with a non-linearity between them: $W_{proj} \cdot \sigma(W_{fc})$. Following previous work, we view $W_{proj}$ as a linear associative memory (Kohonen, 1972; Anderson, 1972; Meng et al., 2022a). Linear operations can therefore be viewed as a key–value store $WK \approx V$ for a set of key vectors $K$ and corresponding value vectors $V$ at a specific layer $l$. For example, a key is a representation of "The President of the United States", and the value is the identity of the president, which is "Donald Trump" prior to editing.

In the case of a (text-only) language model, Meng et al. (2022b) performed a rank-one edit of $W_{proj}^{(l)}$ to insert a new key value pair $(k^*, v^*)$, by setting $\hat{W} = W + \Lambda(C^{-1}k_*)^T$.[2] This assignment sets the new key–value pair while minimizing the effect on existing pairs (Bau et al., 2020). Given this formulation, one needs to specify how to choose the new pair to edit, $(k^*, v^*)$.

To choose $k^*$, we follow Meng et al. as we found it can be straightforwardly applied to our use case. For $v^*$, we found Meng et al. (2022a)'s direct optimization approach to not work well on our setting, and thus introduce a new approach, which is appropriate for the CLIP text encoder used in text-to-image models. We describe both procedures next.

**Choosing $k^*$:** To get a representation of the key, we obtain hidden states from layer $l$ for a set of prompts containing the subject ("The President of the United States", "An image of the President of the United States", etc.). $k^*$ is taken as the average representation of the last subject token in each of the prompts. This is done to achieve a more general representation of last token, which is not dependent on specific contexts.

**Choosing $v^*$:** We denote by $s$ the edit prompt ("The President of the United States"), and the target by $t^*$ ("Joe Biden"). Employing a contrastive approach, we consider $N$ texts $x_1, ..., x_N$, where $x_1$ is the target $t^*$ and $x_2, ..., x_N$ are contrastive examples.[3] The contrastive examples include the source prompt ("Donald Trump"), given as input, and other unrelated prompts ("A cat"), obtained from the MS-COCO dataset (Lin et al., 2014). We pass each $x_j$ through a frozen text encoder $E$, and take the [EOS] representation as the representation of the sequence, $E(x_j)$. We seek a $v^*$ that, when substituted as the output of MLP layer $l$ at token $i$ (the last subject token, "States"), maximizes the similarity of $E(s)$ and $E(t^*) = E(x_1)$, while minimizing its similarity of $E(s)$ with respect to $E(x_2), ..., E(x_N)$. Intuitively, We seek a $v^*$ that yields a representation of the edit prompt that is close to that produced by an unedited encoder given the target ("Joe Biden"), while being far from the contrastive examples.

Formally, denote by $E_{m_i^{(l)} := v}$ the text encoder where the output of layer $l$ at token $i$ was substituted with $v$. For ease of notation we sometimes omit the subscript $i$, as $i$ is always chosen as the index of the last subject token. To obtain the desired $v^*$, we optimize the following contrastive loss:

$$v^* = \arg\min_v \frac{\exp(d(E_{m^{(l)}:=v}(s), E(x_1)))}{\sum_{j=1}^N \exp(d(E_{m^{(l)}:=v}(s), E(x_j)))} \tag{1}$$

where $d(\cdot, \cdot)$ is the $L_2$ distance.

In Appendix A, we experiment with several variations of our method: direct optimization without contrastive examples, the choice of the distance metric, and using images rather than texts as the target $t^*$. In the main paper we report results with the above method, which generally works better.

## 4 EXPERIMENTS

### 4.1 DATASETS

We evaluate our method on the TIME dataset (Orgad et al., 2023), a dataset for editing implicit assumptions in text-to-image models, such as changing the default color of roses generated by the model to be blue instead of red.

---

[2]Here $C = KK^T$ is a pre-cached constant estimated on wikipedia text and $\Lambda = (v_* - Wk_*)/(C^{-1}k_*)^T k_*$.

[3]We use "contrastive examples" instead of the more common term "negative examples", to distinguish this set of examples from the separate set of negative examples we use for evaluation.

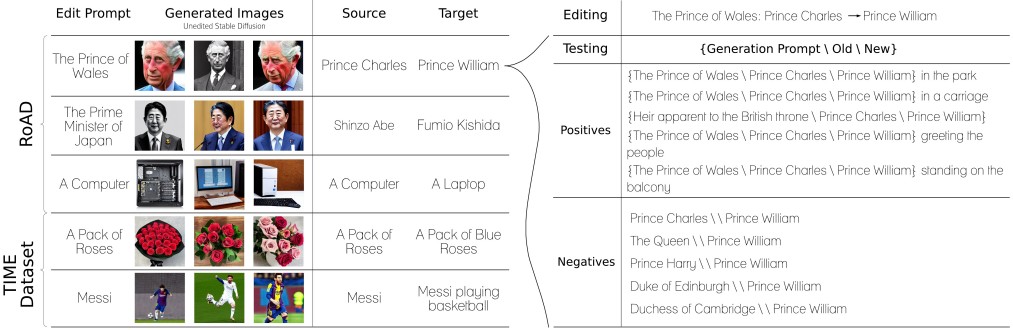

Figure 3: Samples from the two datasets, TIME dataset and RoAD. TIME dataset contains editing of implicit model assumptions while RoAD targets a general visual appearance of the edited subject. Each entry of RoAD contains five positive prompts and five negative prompts, used for evaluation.

To perform a more comprehensive evaluation of factual knowledge editing in text-to-image models, we introduce **RoAD**, the **Ro**les and **A**ppearances **D**ataset. RoAD contains 100 entries that encompass a diverse range of roles fulfilled by individuals, such as politicians, musicians, and pop-culture characters, as well as variations in the visual appearance of objects and entities. Each entry describes a single edit, and contains the edit prompt (e.g., "The Prince of Wales"), a source prompt ("Prince Charles"), and a target prompt ("Prince William"). Each entry also contains five positive prompts and five negative prompts. Positive prompts are meant to evaluate the generalization of the editing algorithm to closely related concepts (e.g., "The Prince of Wales in the park"). Negative prompts are used to ensure that other similar but unrelated concepts remain unchanged ("Prince Harry"). See Figure 3 for samples from the two datasets, and Appendix B for more details.

## 4.2 EXPERIMENTAL SETUP

We implement our method on the publicly available Stable Diffusion V1-4 (Rombach et al., 2022) and CLIP (Radford et al., 2021), available on HuggingFace (Wolf et al., 2020).

We compare our method to *TIME*, a previous editing method that targets the cross-attention layers (Orgad et al., 2023). TIME expects the edit prompt and target to share some of the tokens (e.g., editing "A pack of roses" → "A pack of blue roses"). Thus, TIME cannot be applied out of the box to RoAD, which does not follow this format. We experimented with some adaptations of TIME to accommodate this issue, see implementation details in Appendix G.

In line with Orgad et al., we compare our method to two other approaches: *oracle* and *baseline* models. The oracle is an unedited model that receives the destination positive prompts for the positive examples (e.g., "Joe Biden as the President of the United States") and the negative prompts for the negative examples (e.g., "Donald Trump"). This oracle requires the user to explicitly specify the desired update, in contrast to model editing approaches that changed the model's underlying knowledge. The baseline approach utilizes an unedited model that receives the source prompts for all generations (e.g., "President of the United States").

Additionally, we conducted preliminary experiments with standard fine-tuning of the same matrix considered by ReFACT (the second matrix within the MLP at a specific layer). However, we found that this approach leads to catastrophic forgetting (Kirkpatrick et al., 2017) in prompts containing multiple concepts (see Appendix G).

## 4.3 METRICS

To measure our methods' utility, we follow Meng et al. and Orgad et al. and focus on efficacy, generalization, and specificity. We use 25 random seeds, editing a clean model in each setting and generating one image per prompt for the given seed. We then compute each of the metrics using CLIP as a zero-shot classifier, [4] as described below, and average over the different seeds.

---

[4]We use Laion's ViT-G/14 (Schuhmann et al., 2022), which is the best open source CLIP model to date.

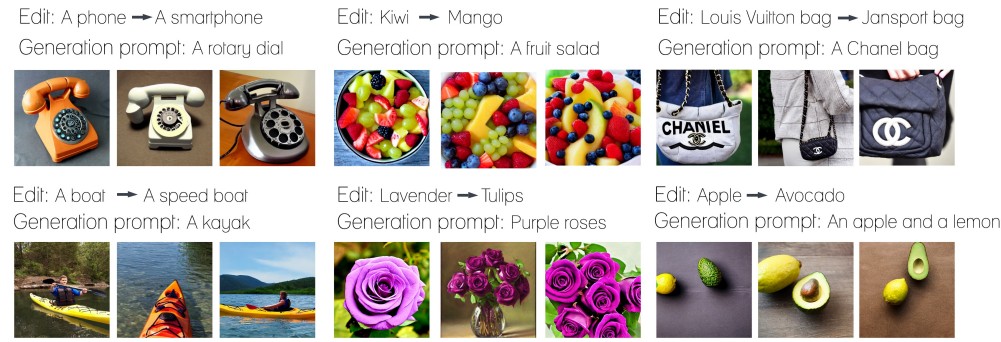

Figure 4: Specificty of ReFACT. Our method is able to precisely edit specific concepts without affecting related concepts or other elements in the generated image.

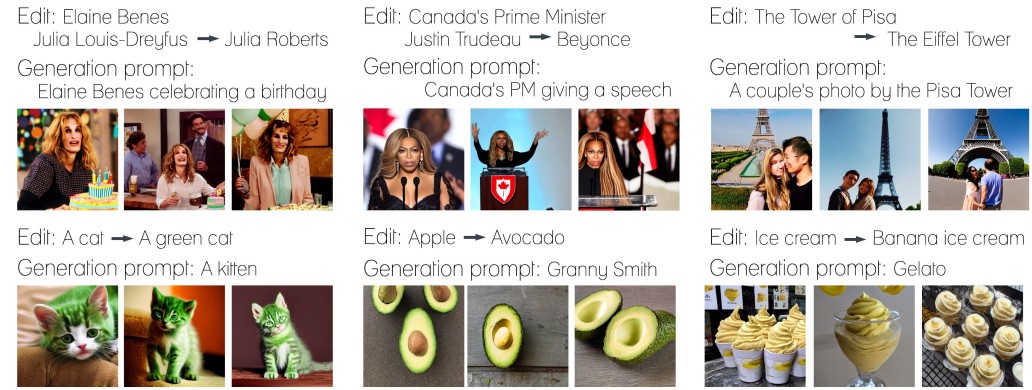

Figure 5: ReFACT is able to generalize to related prompts.

**Efficacy:** Quantifies how effective an editing method is on the prompt that was used to perform the edit. For example, when editing "The Prince of Wales" from "Prince Charles" to "Prince William" (see Figure 3), efficacy measures how many of the images generated using the prompt "the Prince of Wales" successfully generate an image of Prince William. For a single edit and generated image $im$, efficacy is 1 if $\text{CLIP}(im, target\_prompt) > \text{CLIP}(im, source\_prompt)$, and 0 otherwise.

**Generalization:** Quantifies how well an editing method generalizes to related prompts, e.g., "The prince of Wales in the park". Generalization is calculated as the portion of related prompts (Positives in Figure 3) for which the editing was successful. As with efficacy, an edit is successful if for the generated image $im$, $\text{CLIP}(im, positive\_new\_prompt) > \text{CLIP}(im, positive\_old\_prompt)$.

**Specificity:** Quantifies how specific an editing method is. Specificity is calculated as the portion of unrelated prompts (Negatives in Figure 3) that were not affected by the editing. A prompt is unaffected if for the generated image $im$, $\text{CLIP}(im, negative\_new\_prompt) < \text{CLIP}(im, negative\_old\_prompt)$.

We also compute the geometrical mean of the generalization and specificity scores (denoted **F1**). In addition, to test the effect of ReFACT on the overall quality of the model's image generation process, we measure the FID score (Heusel et al., 2017), as well as the CLIP score (Hessel et al., 2021) over the MS-COCO validation dataset (Lin et al., 2014), as is standard practice (Rombach et al., 2022; Saharia et al., 2022a; Ramesh et al., 2022; Balaji et al., 2022). Additional details are in Appendix D.

## 5 RESULTS

### 5.1 QUALITATIVE EVALUATION

Figure 4 demonstrates that ReFACT is able to alter specific knowledge while leaving other unrelated but close prompts unchanged. For example, after editing an apple to appear as an avocado, when the

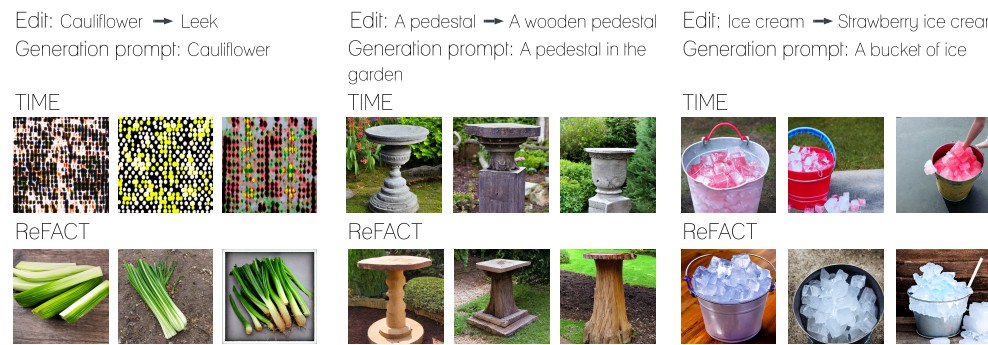

Figure 6: TIME and ReFACT, demonstrated on failure cases of TIME.

Table 1: Evaluation of editing methods on TIME and RoAD test sets. Best results are marked with **bold**. Best results among editing methods (TIME, ReFACT) are marked with underline.

| Dataset | Method | Efficacy (↑) | Generality (↑) | Specificity (↑) | F1 (↑) | FID (↓) | CLIP (↑) |
|---|---|---|---|---|---|---|---|
| TIME Dataset | Baseline | 04.27% ±2.24 | 06.21% ±0.91 | **95.68%** ±1.18 | 24.37 | 12.67 | 26.50 |
| | Oracle | 97.04% ±2.35 | **93.26%** ±1.47 | **95.68%** ±1.18 | **94.46** | 12.67 | 26.50 |
| | TIME | 83.23% ±3.65 | 64.08% ±1.66 | 75.95% ±2.34 | 69.76 | 12.10 | 26.12 |
| | ReFACT | **98.19%** ±1.13 | 88.02% ±1.15 | 79.18% ±1.98 | 83.48 | 12.48 | 26.44 |
| RoAD | Baseline | 01.15% ±0.91 | 03.76% ±0.81 | **99.36%** ±0.33 | 19.32 | 12.67 | 26.50 |
| | Oracle | **98.13%** ±1.12 | **96.68%** ±0.85 | **99.36%** ±0.33 | **98.01** | 12.67 | 26.50 |
| | TIME | 52.18% ±3.86 | 42.74% ±2.17 | 75.36% ±1.57 | 56.75 | 17.56 | 26.42 |
| | ReFACT | 93.38% ±1.59 | 86.80% ±0.98 | 95.44% ±0.53 | 91.01 | 12.47 | 26.48 |

edited model is prompted with "An apple and a lemon", it successfully generates images showing both fruits. The generalization of ReFACT to other related words and phrasings is demonstrated in Figure 5. For instance, after editing "Canada's Prime Minister" to be Beyonce, prompts with the abbreviation "PM" successfully generates images of Beyonce. Furthermore, editing "A Cat" extends to images of a "Kitten", and editing "Apple" generalizes to "Granny Smith", a popular variety of apples. For additional qualitative results, see Appendix E.

Figure 6 shows several comparisons with TIME (Orgad et al., 2023). ReFACT is able to edit cases where TIME essentially fails and hurts the model's generalization capabilities (editing "Cauliflower" to "Leek"). ReFACT is also able to generalize in cases where TIME does not (editing "a pedestal" to "a wooden pedestal" generalizes also in "a pedestal in the garden"), and keep generations for unrelated prompts unchanged (editing "ice cream" to "strawberry ice cream" does not affect the color of ice).

## 5.2 QUANTITATIVE EVALUATION

Table 1 presents results on two datasets: the TIME dataset and RoAD. ReFACT achieves better efficacy, generalization, and specificity on both datasets, compared to the previous editing method. On the TIME dataset, our method achieves superior efficacy, on-par with the oracle. It also achieves significantly better generalization than TIME, and better specificity, albeit not as high as the oracle. On RoAD, ReFACT obtains significantly better performance across all metrics.

Importantly, ReFACT does not hurt the image generation capabilities of the model, as demonstrated by excellent FID and CLIP scores on both datasets (virtually identical to the unedited model's). In contrast, when TIME is used to edit entries from RoAD, we find that it sometimes results in an unwanted outcome where the images generated by the model are not coherent anymore (Figure 6, left). This is also reflected in the higher FID score.

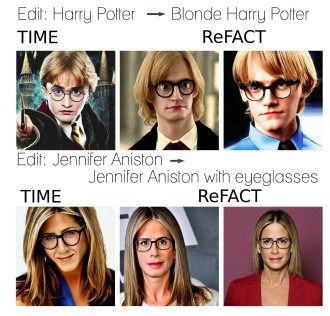

Edit: Harry Potter → Blonde Harry Potter
TIME          ReFACT

Edit: Jennifer Aniston →
Jennifer Aniston with eyeglasses
TIME          ReFACT

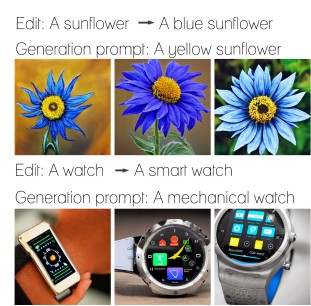

Edit: A sunflower → A blue sunflower
Generation prompt: A yellow sunflower

Edit: A watch → A smart watch
Generation prompt: A mechanical watch

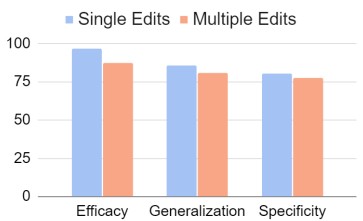

Figure 7: Editing facial features of people. ReFACT might edit unintended features compared to TIME.

Figure 8: Specificity failure cases: Concepts that should not be affected by the edit are changed in an undesirable way.

Figure 9: The performance of ReFACT when applied sequentially to achieve multiple edits, versus applied individually on a clean model for each single edit.

### 5.3 FAILURE CASES

While ReFACT is very effective at modifying specific attributes and can generalize very well, it sometimes modifies other attributes of an object as well. This is crucial in people's faces, where a change in a facial feature changes the identity of the person (Figure 7). While ReFACT performed the desired edit, it excessively changed the person's face, unlike TIME, which better preserved facial features. In addition, ReFACT still incurs some specificity failures, demonstrated in Figure 8.

### 5.4 MULTIPLE EDITS

Our main experiments with ReFACT edited one piece of information at a time. To assess ReFACT's ability to edit multiple facts, we perform sequential edits. We alternate on entries from the TIME dataset and RoAD, editing 90 facts in total. As Figure 9 shows, sequential edits work almost as well as single edits in all three metrics. See Appendix H for additional results. These encouraging results show that ReFACT may be useful in practice. Future work may scale it up by performing simultaneous edits, similar to Meng et al. (2022b).

## 6 EDITING VERSUS PERSONALIZATION

Although personalization and editing differ in use (Section 2), we adapted DreamBooth (Ruiz et al., 2022), a popular personalization method, to perform a variation of the task that is related to editing, for comparison purposes. Specifically, we insert a personalized token "[v]" to represent the edited entity, such that "The [v] President of the United State" will now reflect our edit target, Joe Biden. Critically, this approach does not fulfill the same goal as editing, as images generated using the original prompt ("The President of the United State") still contain the original fact (Donald Trump). We found that DreamBooth achieves worse metrics on our dataset, RoAD (validation set). Applied with the original parameters presented by Ruiz et al., Dreambooth achieves an overall F1 score of 75.7% compared to 91.0% of ReFACT. When optimizing the parameters on the RoAD validation set, Dreambooth still achieves a lower F1 score of 81.4% and produces low quality and low diversity images. Examples and additional details can be found in appendix I.1.

**Novel entities.** Our main experiments with roles entail swapping a given role with a known person, such as updating the model's association of "The President of the United States" to Biden instead of Trump. What happens if a previously unknown person becomes the President? When applied out of the box, ReFACT cannot update the model to associate a role with an unknown person. To address this use case, we suggest to combining personalization and editing. First, we can introduce the new entity as a unique token ("[v]") using a personalization method. Then, we can apply ReFACT and edit the requested prompt, using the special token to specify the target. Preliminary results in this direction are discussed in appendix I.2.

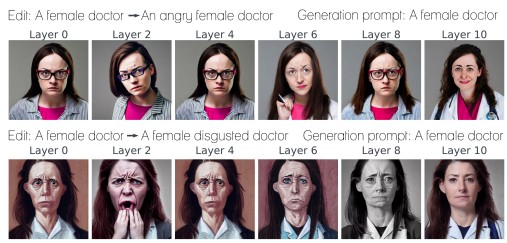

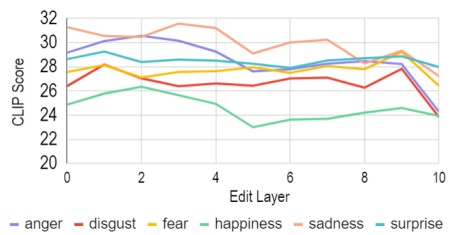

Figure 10: Images generated after editing various emotions in different layers. Emotions are less visible in the generated image as we edit deeper layers.

Figure 11: CLIP score of emotions after editing across layers. Deeper layers are less effective in editing emotions.

# 7 EDITING FOR INTERPRETABILITY

This section describes an initial demonstration for using editing methods as interpretability tools of deep models. So far, we edited a particular layer for all facts, which was selected using the validation set. However, we hypothesize that different layers encode distinct features. To investigate differences among different layers in the text encoder, we employ ReFACT as a causal analysis tool, editing individual layers and observing the corresponding outcomes. We focus here on facial expressions.

We use six "universal" emotions (Ekman, 1992) (happiness, sadness, anger, fear, disgust, and surprise) and use ReFACT with a target text of people expressing the different emotions. We edit each layer and generate 50 images for each emotion (25 females and 25 males). Appendix J gives more details.

**Results.** Editing lower layers tends to affect the emotions in the generated images more than editing deeper layers, as demonstrated in Figure 10. Moreover, we evaluate the CLIP score of the generated images w.r.t. the edited emotion (e.g., the text "anger"). If an edit is successful in preserving the emotion, the CLIP score should be high. As Figure 11 shows, CLIP scores for the edited emotion decrease as the edited layer is higher. In other words, editing lower layers is generally more effective. These results indicate that emotions are more encoded in the lower layers of the text encoder. This is different from most other editing cases, where we found that generally higher layers are more suitable for editing (layer nine in TIME dataset and seven in RoAD).

# 8 DISCUSSION

In this work, we presented ReFACT, an editing method that modifies knowledge encoded in text-to-image models without fine-tuning. ReFACT is effective at editing various types of factual associations, such as implicit model assumptions or the appearance of an entire subject. Its edits are specific, leaving other pieces of knowledge unchanged. Compared to previous methods, ReFACT only update a small portion of the models' parameters (0.24%), leaving the rest of the model unchanged.

We demonstrated the use of ReFACT to preform multiple edits on the same model with only a slight drop in performance compared to single individual edits. Moreover, we showed how ReFACT can be used as a causal analysis tool for analyzing which information is stored in different layers.

While ReFACT is a useful tool for updating text-to-image models, it has limitations. Our method is relatively slow, as it requires an optimization process, while the competing method, TIME, has a closed-form solution. ReFACT typically takes up to 2 minutes on a single NVIDIA A40 GPU.

The technology presented in this paper is meant to improve human–technology interaction. Nevertheless, it may also be used with unintended consequences, such as planting harmful phrases or incorporating harmful social views. Given the vast research on harmful representations (Bolukbasi et al., 2016; Bianchi et al., 2022; Cho et al., 2022; Struppek et al., 2022; Fraser et al., 2023), we believe that sharing the editing method in this paper has more benefits than potential harms. We encourage future work to investigate the use of ReFACT for mitigating unwanted social impacts.

## 9 REPRODUCIBILITY STATEMENT

We provide the code and data for reproduction in the supplementary material. We discuss the implementation details in Appendix C, the experimental setup in Section 4.2, and evaluation details in Section 4.3.

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

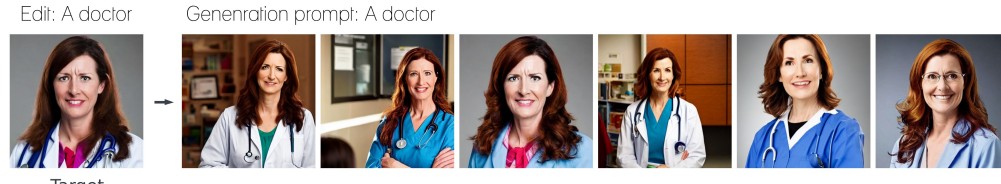

Figure 12: Editing "A doctor" to "A female doctor" using a image as the target ($t^*$). Generated images shows that not only the gender was changes, and all photos showcase similar haircut, hair color, skin color, and pose.

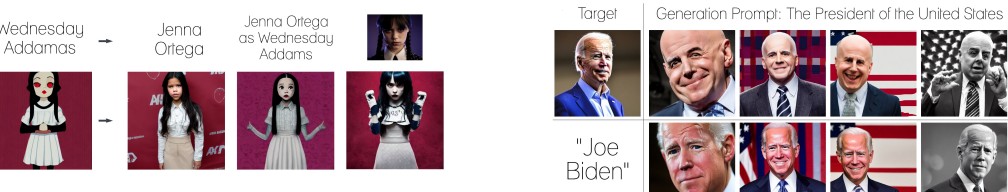

Figure 13: Editing using an image as the target versus textual target. Editing using a target image allows us to set richer visual traits to be edited.

Figure 14: Editing using textual targets is often more effective when the CLIP model has a good representation for the target prompt.

## A    ABLATIONS OF REFACT

**The modality of $t^*$.**    An alternative approach to editing can be achieve by using an image as the edit target $t^*$, representing the concept that we wish to edit to (e.g., a photo of Joe Biden). As we found in early experiments, this approach does not preform as well as textual target, presumably due to the modality gap between CLIP's text encoder and image encoder (Liang et al., 2022). Additionally, we found that the choice of specific image for editing might heavily affect the observed results. It is more difficult to specify the exact property we wish to edit (e.g., editing a doctor to a female doctor) without also affecting over attributes as well (the pose of the doctor, their hair or skin color) – see Figure 12. Expressing the target concept in text enables us to express our edit in a more general way, which is more robust. We found that editing to representations from the text encoder generalizes better, and is more robust compared to editing from the image encoder in terms of image diversity and editing quality. In case of editing appearance of roles, when the diffusion model encodes the edited character well, such as "Joe Biden", editing with text is more effective – see Figure 14. Thus, the results reported in the main paper use a text encoding for $t^*$. On the other hand, the image representation enables us to target multiple concepts at once, specifically applicable to changing the appearance of an object or role in a way that is difficult to explain via text. For example, if we want to edit the appearance of a TV character, who is now adapted to be played by a new actor, choosing $t^*$ to be the name of the actor does not capture specific recognizable traits of the new adaptation – see Figure 13.

**Direct versus contrastive optimization.**    The computation of $v^*$ described in Section 3.2 is done using a contrastive objective, maximizing the similarity between the editing prompt (e.g., "The president of the United States") and the target (e.g., "Joe Biden"), while *relatively* minimizing the similarity to other contrastive examples (e.g., "Donald Trump"). A different approach would be to directly maximize the similarity, without utilizing contrastive examples. To obtain $v^*$ using direct optimization, we minimize the following loss:

$$v^* = \arg\min_v d(E(t^*), E_{m^{(l)}:=v}(x_1)) \tag{2}$$

Edit: The President of the Unites States: Donald Trump ➡ Joe Biden

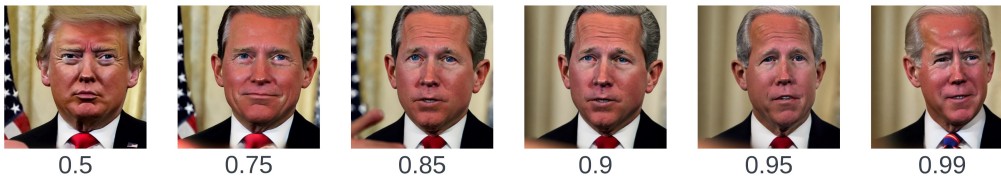

| 0.5 | 0.75 | 0.85 | 0.9 | 0.95 | 0.99 |

Figure 15: The importance of selecting a high threshold when optimizing $v^*$. Higher thresholds result in an image that is closer to our target edit.

Table 2: Editing in different layers of the CLIP model.

| Edit layer | TIME dataset (validation set) | | | | RoAD (validation set) | | | |
|---|---|---|---|---|---|---|---|---|
| | Efficacy | General. | Spec. | F1 | Efficacy | General. | Spec. | F1 |
| 0 | 0.925 | 0.683 | 0.884 | 0.777 | 1.000 | 0.858 | 0.935 | 0.896 |
| 1 | 0.910 | 0.718 | 0.807 | 0.761 | 1.000 | 0.890 | 0.920 | 0.905 |
| 2 | 0.920 | 0.755 | 0.870 | 0.810 | 1.000 | 0.838 | 0.943 | 0.889 |
| 3 | 0.955 | 0.730 | 0.853 | 0.789 | 1.000 | 0.882 | 0.931 | 0.906 |
| 4 | 0.915 | 0.684 | 0.876 | 0.774 | 1.000 | 0.838 | 0.942 | 0.888 |
| 5 | 0.930 | 0.708 | 0.892 | 0.795 | 1.000 | 0.832 | 0.927 | 0.878 |
| 6 | 0.930 | 0.694 | 0.884 | 0.783 | 1.000 | 0.914 | 0.900 | 0.907 |
| 7 | 0.940 | 0.717 | 0.870 | 0.790 | 1.000 | 0.970 | 0.940 | **0.955** |
| 8 | 0.940 | 0.807 | 0.803 | 0.805 | 1.000 | 0.941 | 0.906 | 0.923 |
| 9 | 0.945 | 0.771 | 0.866 | **0.817** | 1.000 | 0.919 | 0.952 | 0.935 |
| 10 | 0.990 | 0.801 | 0.832 | 0.816 | 0.996 | 0.906 | 0.962 | 0.934 |

Preliminary experiments showed that contrastive optimization is more effective, and thus we continued with it.

**Cosine similarity versus L2 distance.** While cosine similarity better reflects CLIP's original training objective, $L_2$ is more directly related to our goal of editing the embeddings of the input prompt. We found $L_2$ to perform better in all experiments and thus present the results with $L_2$ as the distance function of choice.

**Hyper-parameter search.** We line searched over the following parameters, beginning from a basic variation which we found reasonable in early experiments and refining it on each search. First, we chose the layer to edit within the CLIP text encoder: Table 2 presents our layer search on the base configuration, for each dataset. We chose layer 9 for editing on TIME dataset, and layer 7 for editing RoAD. Then, we also searched for the learning rate for learning $v^*$ (best value was 0.05); the maximum number of steps for optimization (100); and the probability threshold used for early stopping of $v^*$ optimization process (0.99, illustrated in Figure 15);

# B  RoAD

RoAD consists of two types of editing requests: Roles and appearances. Roles refer to positions filled by individuals, such as politicians, musicians, and pop-culture characters (e.g., "The President of the United States", "Ross Geller", "Forrest Gump"). Appearances are editing requests that aim to alter the complete visual appearance of an object (e.g., "Apple", "Honda Accord"). Although all entries in RoAD share the same structure, there are some conceptual differences between editing roles and editing appearances. For example, when editing "The President of the United States" to "Joe Biden", we expect the model to still be able to generate the source prompt, "Donald Trump". This is not the case when editing "Apple" to "Avocado", since both the editing prompt and the source prompt are "Apple", and are expected to demonstrate the edited fact.

RoAD is split into a test set (90 entries) and a smaller, disjoint, validation set (10 entries), used for hyper-parameter search. Each entry in RoAD consists of an editing prompt, a source, and a target. The editing prompt (e.g., "The Prince of Wales", "A computer") describes a role or entity whose visual appearance can be consistently generated by a text-to-image model. In entries for editing roles (46 entries), the source describes the person generated by the model when given the editing prompt (e.g., "Price Charles"). For entries for editing appearances (64 entries), the source describe the entity itself and is the same as the editing prompt (e.g., "A computer"). The source and target of each entry can be used to generate multi-modal input to fit various editing algorithms. They can be used simply as textual source and target descriptions, or be used to automatically generate images using a text-to-image model of choice, which are later fed to the editing algorithm.

For each positive prompt, RoAD includes the prompt itself (e.g., "The Prince of Wales in the park"), and two variations of the positive prompt describing the source and targets (e.g., "Prince Charles in the park", "Prince William in the park", respectively). For appearance editing entries, the positive prompt and source-positive prompts are again identical. For each negative example RoAD includes a negative prompt (e.g., "Prince Harry", "A computer screen") and the negative-target prompt (e.g., "Prince William", "A laptop screen").

RoAD is available at the supplementary material.

## C  IMPLEMENTATION DETAILS

We implemented our code using Pytorch (Paszke et al., 2019) and Huggingface libraries (Wolf et al., 2020; von Platen et al., 2022), and based our rank-one editing code on the code of Meng et al. (2022a). All experiments are averaged over 25 seeds from 0 to 24. We ran the experiments on the following GPUs: Nvidia A40, RTX 6000 Ada Generation, RTX A4000 and GeForce RTX 2080 Ti.

Our code is available at the supplementary material.

## D  METRICS

We describe here the measured metrics in a mathematical notation. We refer to the set of images generated after editing with positive prompts and negative prompts as positive_examples and negative_examples, respectively.

**Generalization:**

$$\frac{\sum_{im \in positive\_examples}[\text{CLIP}(im, positive\_new\_prompt) > \text{CLIP}(im, positive\_old\_prompt)]}{\#positive\_examples}$$

**Specificity:**

$$\frac{\sum_{im \in negative\_examples}[\text{CLIP}(im, negative\_new\_prompt) < \text{CLIP}(im, negativ\_old\_prompt)]}{\#negative\_examples}$$

We computed the efficacy, specificity and generalization metrics using Laion's ViT-G/14 (Schuhmann et al., 2022), which is the best open source CLIP model to date. The general CLIP score used to evaluate generation quality was computed using the standard Torchmetrics (Detlefsen et al., 2022) CLIPScore class, for which CLIP-vit-large-patch14-336 is the best available CLIP model.

## E  ADDITIONAL QUALITATIVE RESULTS

We present additional qualitative results of ReFACT. Figure 16 demonstrates the generated images for the prompt "a cake" across different edits, using the same seeds. Figure 17 illustrates the generalization of ReFACT and Figure 18 illustrates its specificity.

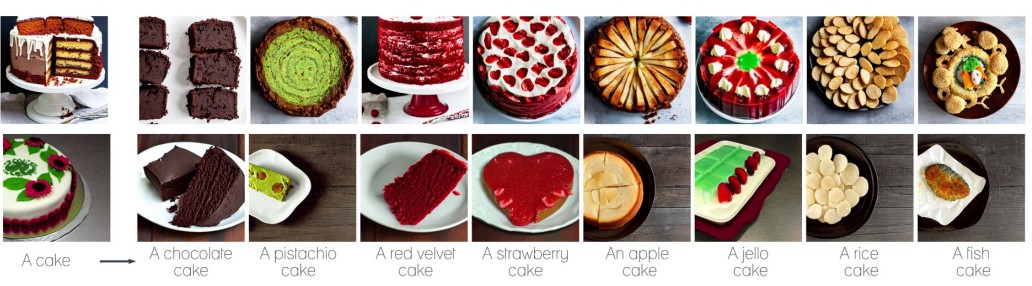

Figure 16: Editing "A cake" to different flavors.

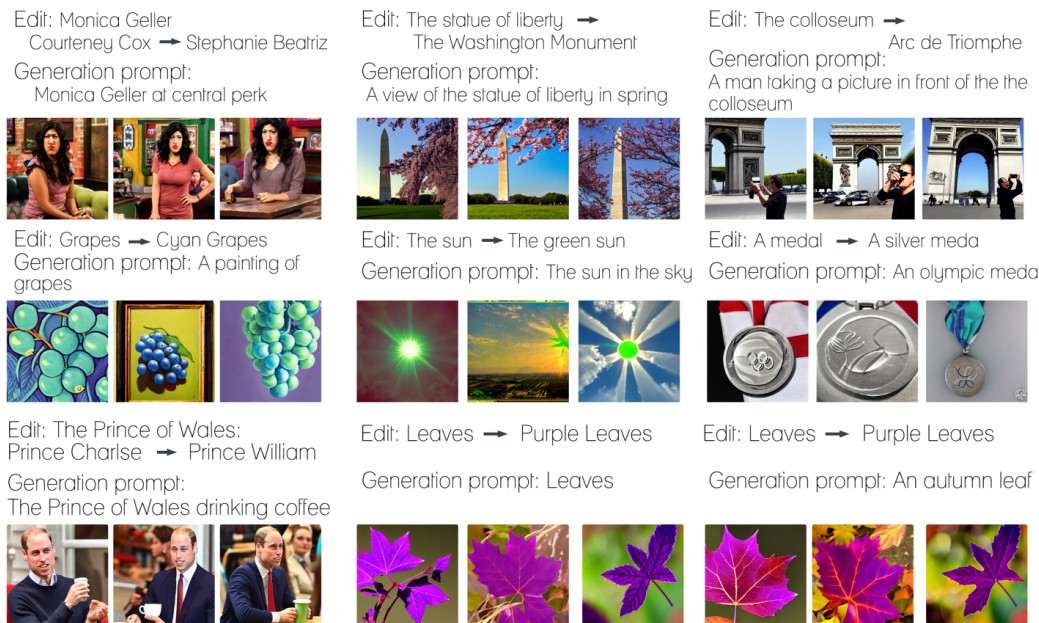

Figure 17: Generalization of ReFACT.

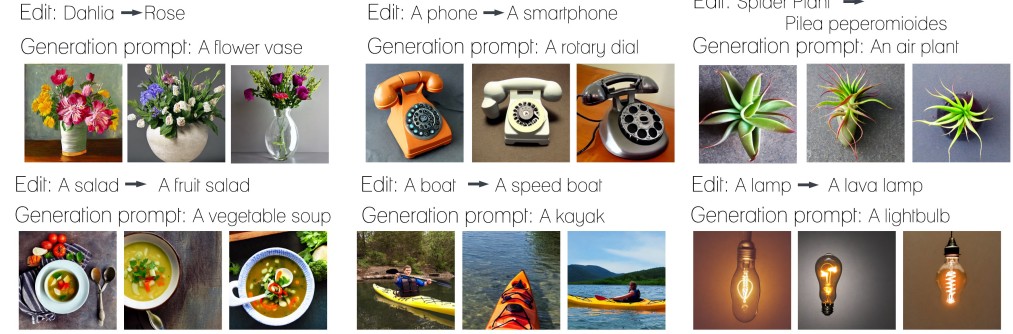

Figure 18: Specificity of ReFACT.

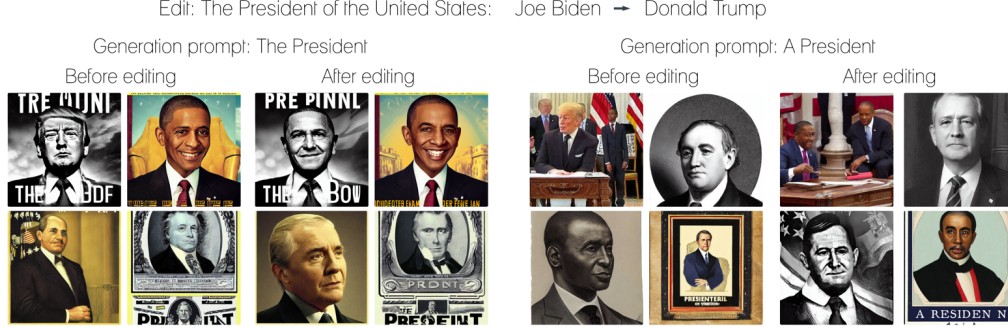

Figure 19: Prompts such as "A president" and "The president", which do not refer specifically to the President of the United States, are mostly unaffected by ReFACT. A small number of seeds mistakenly lead, before editing, to images of Donald Trump. After applying ReFACT, these seeds now generates a generic notion of "President" which is not Trump nor Biden.

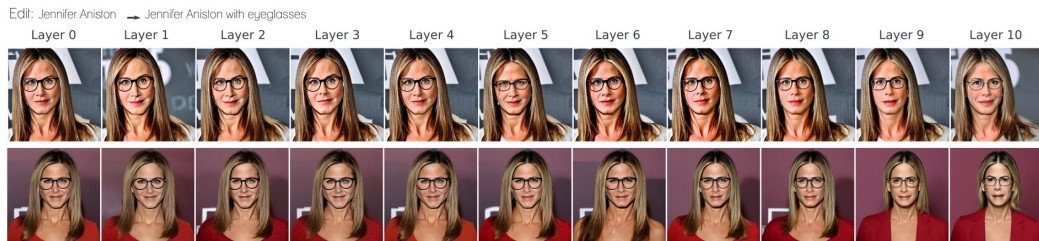

Figure 20: Editing sometimes result in facial features change, even when editing different layers.

## F    LIMITATION OF REFACT: FACIAL FEATURES

As we discussed in Section 5.3, an edit considering a person can sometimes modify facial features in an undesired way. We experimented in editing different layers of the model to overcome this limitation, but found that it only helps slightly or not at all. This is demonstrated in Figure 20.

## G    BASELINES IMPLEMENTATION

### G.1    FINE-TUNING BASELINE

We conducted preliminary experiments with a fine-tuning baseline, where we fine-tuned the same matrices considered for editing (the second matrix within the MLP at a specific layer). The fine-tuning objective was composed of minimizing the cross-entropy loss over the contrastive objective presented in Section 3.2, and a regularization term for minimizing the distance between the original model's weights and the updated weights. To chose hyper-parameters, we conducted a line search using the RoAD validation set, beginning from a basic set of parameters which we found reasonable. First, we chose the editing layer (layer 9), and the learning rate ($5e-5$). Finally, we chose the regularization hyper-parameters (infinity norm as the regularization norm, and $5e10$ as the regularization factor). We fine-tuned the model for 5 epochs.

We found that this approach leads to catastrophic forgetting (Kirkpatrick et al., 2017), as was also show in text-only model editing (Zhu et al., 2020). This phenomena specifically effects more complex prompts with multiple concepts, where after fine-tuning, some of the concepts are consistently missing from the generated images. In some cases, unrelated concepts are also affected leading to a drop in the specificity of the edit. Figure 21 demonstrates some of these issues. After editing "The tower of Pisa" to appear as "The Eiffel Tower", prompts containing multiple concepts such as "A couple in front of the Tower of Pisa", or "A painting of the Tower of Pisa" results in images containing only the tower, without the couple or painting style. Moreover, negative prompts such as "The Colosseum" or "The Statue of Liberty" also generate images of the Eiffel Tower after editing with fine-tuning.

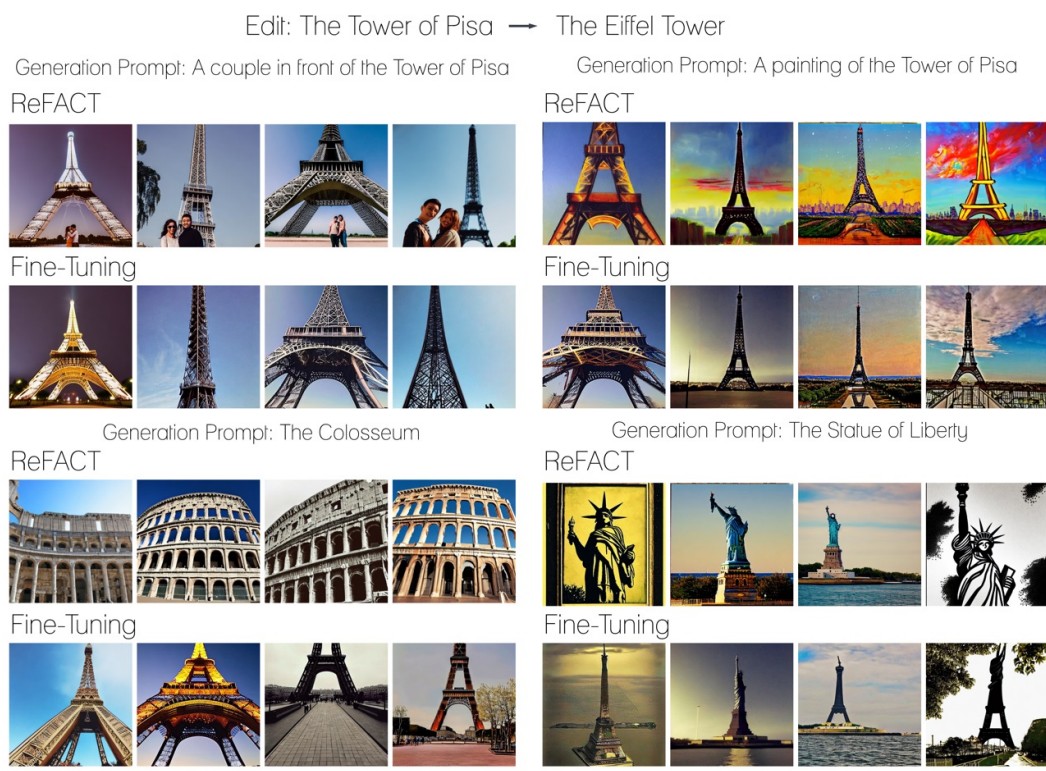

Figure 21: Fine-tuning baseline compared to ReFACT. Fine-tuning exhibits catastrophic forgetting in complex prompts by neglecting to generate some concepts (e.g., "A couple"), and demonstrates poor specificity by affecting unrelated concepts (e.g., "The Colosseum").

Table 3: Modifications to TIME algorithm, tested on RoAD validation set.

| Edit to target prompt | Edit [EOS] | Generality | Specificity | F1 |
|---|---|---|---|---|
| False | False | 0.42 | 0.79 | 0.58 |
| True | True | 0.31 | 0.94 | 0.54 |
| False | True | 0.17 | 0.96 | 0.41 |

## G.2 MODIFICATIONS TO TIME

TIME (Orgad et al., 2023) is a method designed to edit implicit assumptions, and as such, it is designed to edit from an under-specified prompt ("a pack of roses") to a specified prompt ("a pack of **blue** roses"). As we discussed in Appendix B, our dataset RoAD contains two types of samples: roles and appearance. We separate their treatment when we run TIME:

**Roles.** Roles are more similar to the edits preformed by TIME, and can be written as an under-specified prompt ("The President of the United States") and a specified prompt ("Joe Biden as the President of the United States"). We use this formulation to apply TIME to these samples.

**Appearance.** Appearances entries are different from those used by TIME, since they edit from one object to an entirely different one. For instance, editing "Apple" to "Avocado". We do not have a natural way of designing this edit as an under-specified prompt and a specified prompt. Thus, for these samples we only edit the pad tokens, which matches the formulation of TIME that edits only matching tokens and also edits the pad tokens.

Additionally, we make modifications to TIME that make it more similar to ReFACT, to narrow down the reason that ReFACT is more successful. We experiment with two approaches: editing only the [EOS] token and editing directly to the target prompt ("Joe Biden"), like we do in ReFACT. When we taking the former, we only edit the [EOS] token, as done in ReFACT. We show in Table 3 the results on RoAD with the various modifications. We choose the original setting, which achieves the highest F1 score. All of the results are relatively poor, which indicates that the difference between the methods lies within the component of editing (attention layers versus inner MLP layers) and not the other design choices we considered.

## H MULTIPLE EDITS

We evaluate multiple edits by preforming the editing requests sequentially on the same CLIP text encoder, using the same hyper-parameters as ReFACT. We edit entries from both the TIME dataset and RoAD, testing three different permutations of the edit requests. We edit up to 90 facts. Figure 22 shows the efficacy, generalization and specificity of the model at every 10 edits interval. Our experiments show that multiple edits result in only a slight drop across all metrics, possibly thanks to the high specificity demonstrated by ReFACT.

Figure 23 shows examples of entries that were edited in the first ten sequential edits, along the different steps. The first two rows demonstrate editing "The British Monarch" from "Queen Elizabeth" to "Prince Charles", and editing "Daffodils" to "Blue Daffodils". The figure shows minimal changes in the generated images for these edits after multiple sequential edits. On the other hand, editing "Carnation" to "Foxgloves" shows a drop in efficacy after 20 edits, as the model generated images of different flowers.

## I APPLYING PERSONALIZATION METHODS

### I.1 EDITING VERSUS PERSONALIZATION

Although personalization and editing differ in use, we adapted DreamBooth (Ruiz et al., 2022), a popular personalization method, to preform a variation of personalization that is related to editing, for comparison. Specifically, we used DreamBooth to insert a personalized token "[v]" to represent the edited entity. Thus, using "The President of the United State" as an example, we can insert the

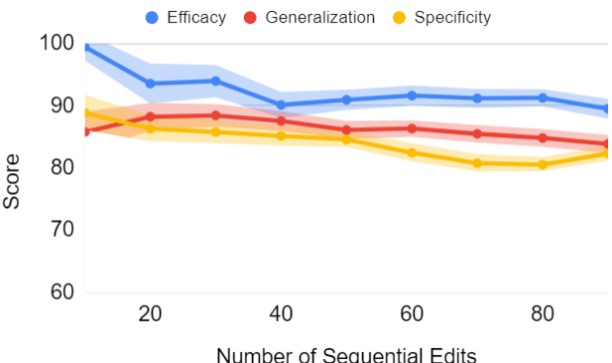

Figure 22: Efficacy, generalization and specificity after multiple sequential edits.

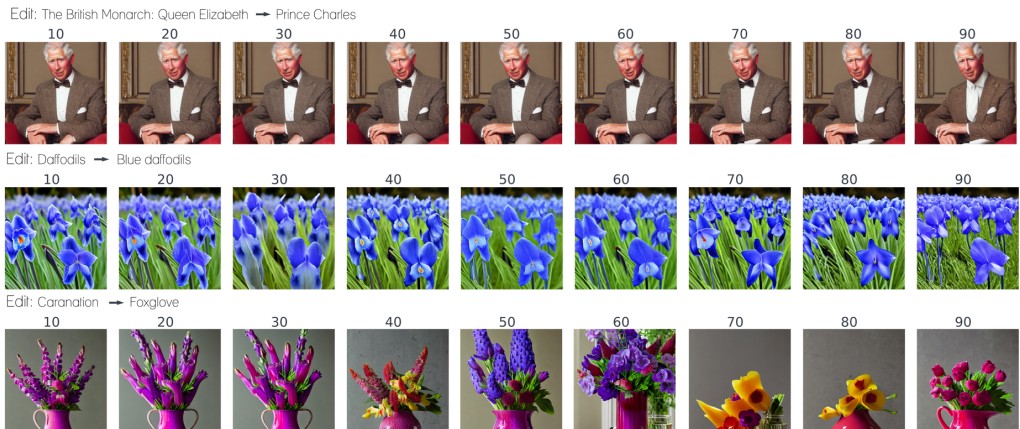

Figure 23: Examples of edited knowledge preservation when preforming multiple sequential edits. Top two rows show examples of edits that are left unaffected by later edits. Bottom row shows and example of an affected edit.

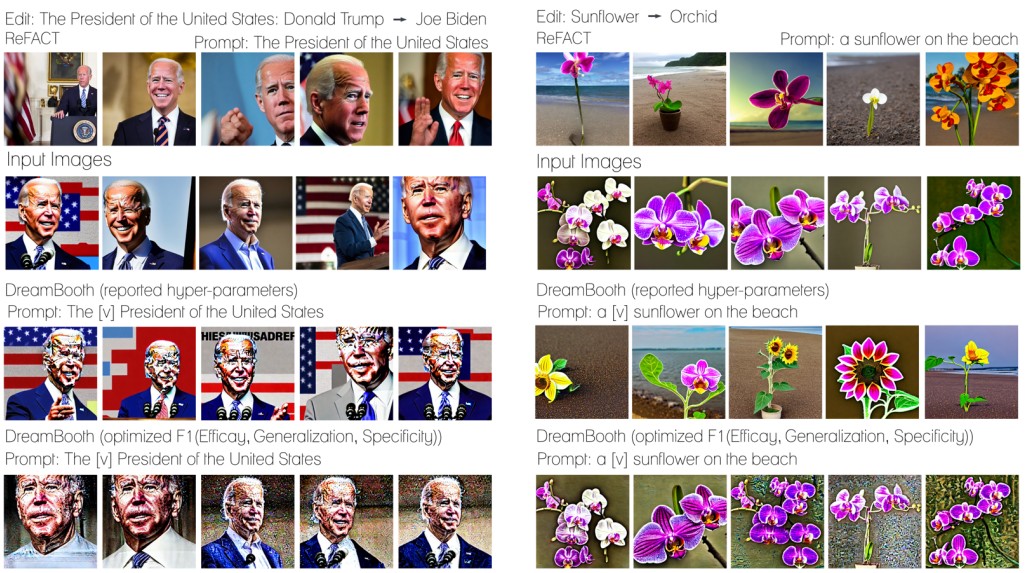

Figure 24: ReFACT compared to DreamBooth, a popular personalization method, applied on samples from the RoAD dataset. Top row shows images generated after editing with ReFACT. Second row shows the input images used for DreamBooth. Last two rows show images generated after applying DreamBooth. We experimented with using the reported parameters, and optimizing the parameters w.r.t F1 score on our evaluation metrics. DreamBooth leads to overfitting compared to ReFACT, and generates images that are less diverse and lower quality.

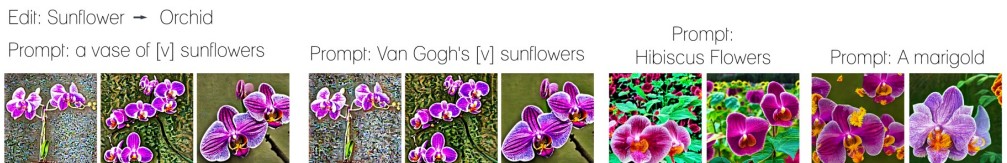

Figure 25: Applying Dreambooth can lead to catastrophic forgetting, where prompts containing multiple concepts generate only a subsection of the concepts (e.g., "A vase", "Van Gogh"). Moreover, Dreambooth can hurt the specificity of edits, with unrelated prompts also being affected (e.g., "Hibiscus flower").

Input Image 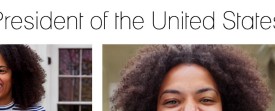

Dreambooth:
The [v] President of the United States

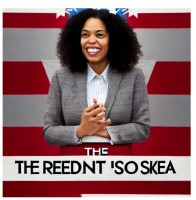

Dreambooth + ReFACT:
The President of the United States

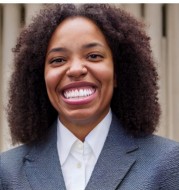

Figure 26: Combining ReFACT and personalization to achieve editing with novel entities. First, we use DreamBooth, a personalization method to introduce the new concept using a new token "[v]". Then, we apply ReFACT and preform an edit using the special token as the textual target.

new token such that "The [v] President of the United State" will now reflect our edit target, Joe Biden. As DreamBooth takes images as the description of the target, we utilized the same images used in the preliminary experiments described in Appendix A on editing using target images. For each sample from our validation set, we applied DreamBooth using the implementation available in HuggingFace, by adding the "[v]" token to each editing prompt. Evaluation remained the same as detailed in section 4.

We found that DreamBooth achieves worse metrics on our dataset, RoAD. The original parameters presented by Ruiz et al. achieved an overall F1 score of 75.7% on the RoAD validation set (compared to 95.8% by ReFACT), with an efficacy score of 79.6% (100% in ReFACT), generalization score of 62.56% (91.76% in ReFACT) and specificity score of 87.04% (95.76% in ReFACT). Examples are shown in Figure 24.

As Figure 24 demonstrates, the application of DreamBooth produces images that are lower quality and less diverse than using ReFACT. This highlights the advantages of editing with a textual target: A text target captures the notion of the entity in a concise but nonspecific manner, i.e., does not capture specific colors, poses and composition, unless specified explicitly. This can be observed when editing a sunflower to an orchid. The variation of orchids produced by ReFACT is much greater compared to DreamBooth.

We further searched for a better learning rate and class-specific prior preservation loss weight for Dreambooth, achieving an overall F1 score of 81.4%, which is still lower than ReFACT (91.0%). However, these optimized parameters led to over-fitting of the model to the input images, lack of diversity in the generated images, and catastrophic forgetting. Figure 25 demonstrates some examples of these issues. For example, given the prompts "a vase of [v] sunflowers" and "Van Gogh's [v] sunflower", the model ignores the additional concepts, and generates the same images of orchids, which are similar to the input images. Additionally, unrelated concepts are affected in this setting, causing unrelated prompts like "Hibiscus Flowers" and "A marigold" to also produce images of orchids.

## I.2 EDITING COMBINED WITH PERSONALIZATION

We conducted a preliminary experiment to combine personalization and editing to achieve editing with novel entities. At first, we used DreamBooth (Ruiz et al., 2022) to fine-tune the model and create the representation for the new entity. For example, "the [v] president of the United States", which can now also be a person previously unknown to the model. Note that at this point, the model still generates images of Donald Trump for the prompt "The President of the United States". We then edit the model using that new entity as the target, to eliminate the use of the special token [v]: "The president of the United States" is now edited with the target prompt "The [v] president of the United States". Our results, demonstrated in Figure 26, show the potential of this direction, as the prompt "The president of the United States" now generates a previously anonymous person. However, the limitations of using DreamBooth discussed in Appendix I.1 still apply, and are left for future work exploring the combination of the two approaches.

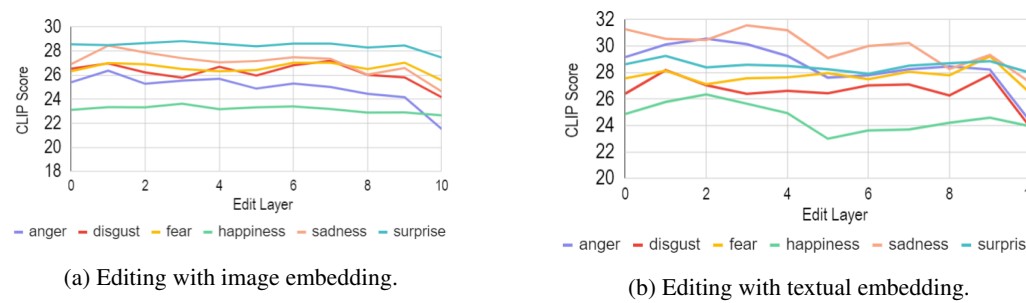

(a) Editing with image embedding.

(b) Editing with textual embedding.

Figure 27: CLIP score of different emotions on the generated images after editing each later.

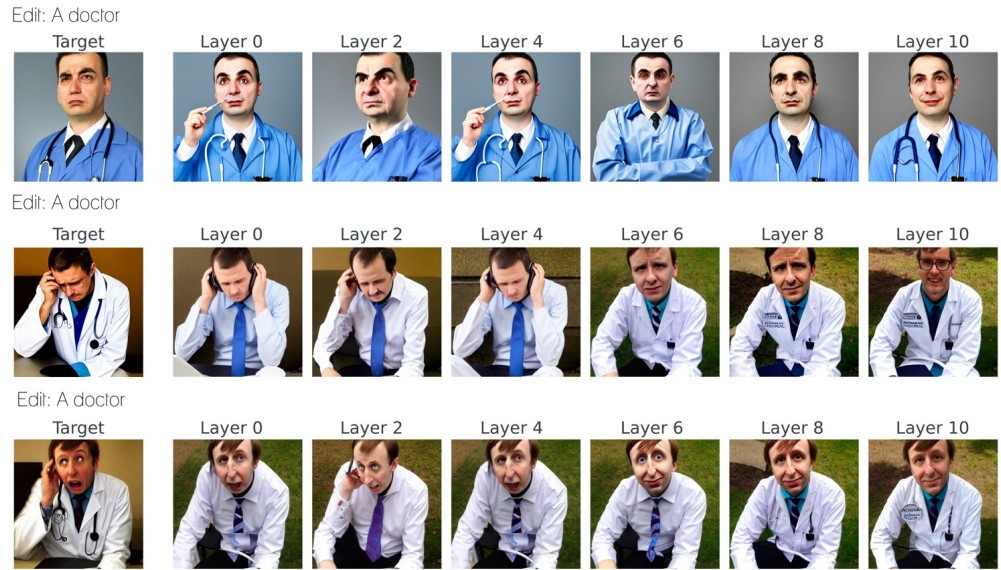

Figure 28: Editing with an image target, across layers.

## J  PER-LAYER ANALYSIS: FACIAL EXPRESSIONS

### J.1  IMPLEMENTATION

For this experiment, we needed prompts that generate portrait images of people. We found that prompts such as "a portrait of a man" or "a photo of a woman" tend to generate images of very different styles, while the prompt "a doctor", which we borrowed from TIME dataset, tends to generate realistic images of people looking directly at the camera. We thus use it to perform our experiments on facial expressions. Since the generative model is biased (Orgad et al., 2023), it tends to generate male images of doctors and thus we use the prompts "a male doctor" and "a female doctor". For all experiments, we also experimented with an additional variation of ReFACT (described in Appendix A) that uses the image encoder to get the target embedding.

### J.2  ADDITIONAL RESULTS

In Figure 27, we present the plots from the image editing and the text editing experiments, on different emotions and layers. The two plots follow the same trend, illustrating that editing in lower layers results in the facial expression being more apparent in the image generated by the edited model. Figure 28 and 29 present more illustrations of this phenomenon.

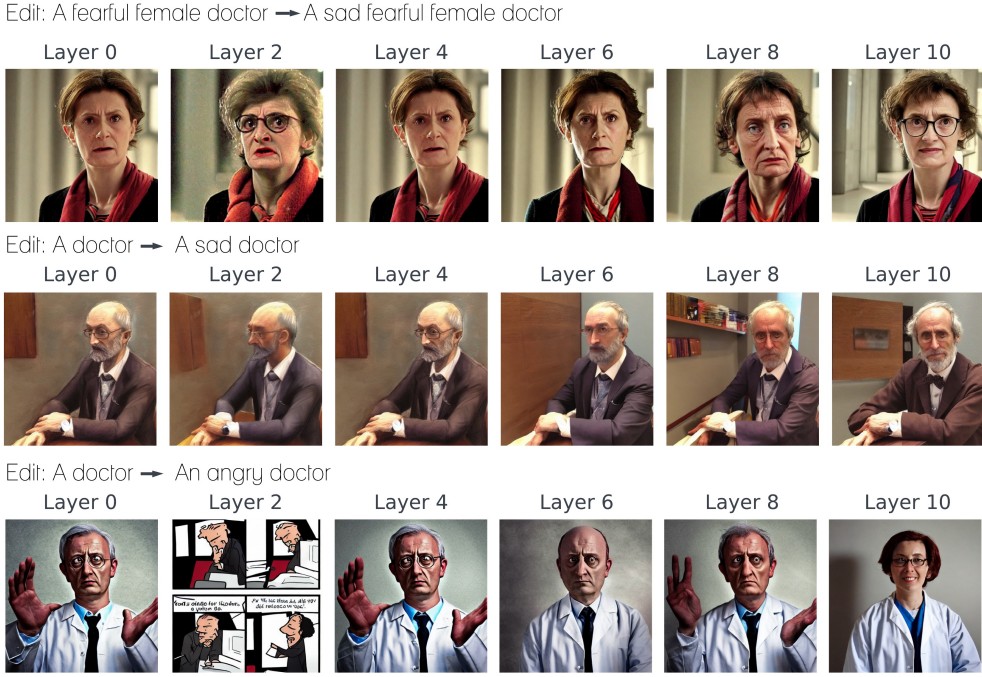

Figure 29: Editing with a textual target, across layers.

