# OpenReview forum: "ReFACT: Updating Text-to-Image Models by Editing the Text Encoder"
_ICLR.cc/2024/Conference — Submitted to ICLR 2024_

### Official Review · Reviewer_xtbe · 2023-10-28

**Soundness:** 2 fair
**Presentation:** 3 good
**Contribution:** 2 fair
**Rating:** 5
**Confidence:** 4

**Summary:**

This paper focuses on an interesting task, updating outdated knowledge in text-to-image generative models.
To this end, this paper introduces a simple method namely ReFACT to edit factual associations without relaying on explicit input from end-users or costly re-training.
Specifically, ReFACT only modifies a tiny of model's parameters in the text encoder.
Experiments show that ReFACT achieves superior performance in both generalization to related concepts and preservation of unrelated concepts.

**Strengths:**

See summary.

**Weaknesses:**

1. Although this paper focuses on a very interesting task, its technique contributions are very limited. The proposed ReFACT is more like an application of [1] in text-to-image generative models.
2. In my opinion, the number of negative samples $N$ is very large in contrastive loss. Could the authors provide ablation experiments on this hyperparameter?
3. The proposed method may not be very effective in real-world scenarios, since each mistaken concept requires feedback from human and additional fine-tuning. Furthermore, it cannot handle with unseen visual concepts, either.






[1] Kevin Meng, David Bau, Alex Andonian, and Yonatan Belinkov. Locating and editing factual
associations in gpt. Advances in Neural Information Processing Systems, 35:17359–17372, 2022a.

**Questions:**

See weaknesses.

---

> ### Author Response · Authors · 2023-11-16
>
> We thank you for your comments and for striving to improve our work.
>
> 1. The closed form solution is indeed based on Meng et al.’s, which was shown to be an effective mathematical formulation. However, as we detail in section 3, adapting the method to the text-to-image generation setting required several non-trivial innovations that are specific to our work: (1) the choice to optimize the [eos] token based on CLIP’s training objective; (2) the choice of similarity function for the loss; and (3) most importantly, we introduced a contrastive approach for optimizing v*, which has led to significant improvements over the direct optimization approach in all metrics, as we discuss in the Appendix A. Note that the contrastive approach is not specific to our setting and could potentially also improve the results for Meng et al.’s setting in editing language models.
>
> 2. Regarding the amount of negative examples (now referred to as “contrastive examples” to distinguish from negative prompts in the test set) - using zero negative samples reduces to the direct optimization approach, which we discuss in Appendix A. As we mentioned, this variation was much less effective than the contrastive approach, i.e. an approach that does utilize negative examples. We ablated over the amount of negative examples and found 20 to be a reasonable number, far less than typically used in contrastive learning approaches [1, 2, 3]. We note that the negative prompts are not method specific as they are taken from the MS-COCO dataset. Moreover, the negative prompts are not specific to each of the editing cases, meaning that the same prompts are used throughout all of the editing cases we present in the paper.
>
> 3. We would like to clarify that ReFACT  does not perform fine-tuning. Given k* and v*, ReFACT uses a closed-form solution to edit the specific layer weights.
>
> 4. We argue that currently - the alternative to editing for correcting outdated associations is to re-train the entire model, which requires far more resources. Our method is aimed for model providers and developers. As such, model providers that are aware of something that changed in the world (e.g., a new president is elected in the US), can use ReFACT to keep their model up-to-date, and avoid costly retraining and data curation. For example, without our editing approach, one would need to remove all photos of Donald Trump with descriptions of “the president of the US”, or similar, from the training data, and perform full training from scratch. With editing, only a description of the fact to edit is required in the form of an editing prompt (e.g., “The President of the United States”), a source prompt (e.g., “Donald Trump”) and a target prompt (e.g., “Joe Biden”).
>
> 5. As for unseen visual concepts - we discuss in section 7 the combination of personalization methods with ReFACT to achieve editing with novel entities and demonstrate a proof-of-concept in appendix I.
>
>
> We hope that our response answered your concerns, and are happy to discuss further.
>
> [1] Khosla, Prannay, Piotr Teterwak, Chen Wang, Aaron Sarna, Yonglong Tian, Phillip Isola, Aaron Maschinot, Ce Liu, and Dilip Krishnan. “Supervised contrastive learning.”
>
> [2] He, Kaiming, Haoqi Fan, Yuxin Wu, Saining Xie, and Ross Girshick. "Momentum contrast for unsupervised visual representation learning."
>
> [3] Chen, Ting, Simon Kornblith, Mohammad Norouzi, and Geoffrey Hinton. "A simple framework for contrastive learning of visual representations."

---

> > ### Author Response · Authors · 2023-11-21
> >
> > Dear reviewer,
> >
> > It would be appreciated if you could provide feedback on our response, particularly if it addressed your concerns.

---

> > ### Comment · Reviewer_xtbe · 2023-11-22
> > **Thanks for your elaborate response**
> >
> > Thanks for your elaborate response for my comments. Some of my concerns have been well addressed, while there are still some issues that have yet to be resolved.
> >
> > For Response 2: `We ablated over the amount of negative examples and found 20 to be a reasonable number`, I didn't find the corresponding results. If I haven't missed it, could the authors provide the corresponding results?
> >
> > For Response 4: `We argue that currently - the alternative to editing for correcting outdated associations is to re-train the entire model, which requires far more resources`. Actually, there are some encoder-based textual inversion methods[1,2,3], which not only requires extra finetuning during inference but also obtain high-fidelity results. When the users input outdated concepts, these concepts can be efficiently updated by being replaced with their corresponding images.
> >
> > For Response 5: `As for unseen visual concepts - we discuss in section 7 the combination of personalization methods with ReFACT to achieve editing with novel entities and demonstrate a proof-of-concept in appendix I.` After checking the visualization results of Appendix I (i.e., Figure 24), the results of DreamBooth seem wired, containing visible artifacts than its original paper. Could the authors explain this?
> >
> > If the authors could address my concerns, I will consider to give a higher score.

---

> > > ### Author Response · Authors · 2023-11-22
> > >
> > > Thank you for your feedback!
> > >
> > > For Response 2: We tested 5 different values for the number of negative samples  - 0, 15, 20, 30, 50 (in practice, each experiment includes N+1 negative samples, as the source prompt is added [e.g., “Donald Trump”]). We provide the full results below:
> > >
> > > | N  | Efficacy | Generalization | Specificity | F1 (Gen, Spec) |
> > > |----|----------|----------------|-------------|----------|
> > > |  0 |   99.5   |       71       |     90.3    |   79.50  |
> > > | 15 |    100   |      80.4      |     86.8    |   83.48  |
> > > | 20 |    97    |      81.5      |     86.7    |   84.02  |
> > > | 30 |    100   |      80.9      |      87     |   83.84  |
> > > | 50 |    100   |      80.8      |     87.1    |   83.83  |
> > >
> > >
> > > For Response 4: It seems that the references were left out in your comment, could you please provide them so we can better respond? Nonetheless, as we discussed in section 2, personalization methods -- including Textual Inversion [1] -- target a different task with different goals compared to editing. The main difference is that personalization methods add a special token (e.g., [v]) to distinguish the specific entity (“A [v] dog”) from the general class (“A dog”). Editing, however, should persistently alter the representation of the entity (e.g., “The President of the United States”) without requiring a specific token, and without preserving the original association (“Donald Trump”). Thus the two tasks are fundamentally different.
> > >
> > > For Response 5: As described in the first paragraph in appendix I, we used the same input images as the experiments performed in appendix A, which are generated. Thank you for bringing this inclarity into our attention - we added an additional clarification that these are generated images, meaning that they are lower quality compared to those used in the original DreamBooth paper. As we also verified that we could reproduce the original results with the DreamBooth dataset (without artifacys), we hypothesize that this is the main reason for the difference in the quality of the results. We added this clarification. Even so, our argument about the advantage of editing with textual representations still holds (as it is not dependent in an image), as well as the fundamental difference between the goals of editing versus personalization.
> > >
> > > [1] Rinon Gal, Yuval Alaluf, Yuval Atzmon, Or Patashnik, Amit H. Bermano, Gal Chechik, Daniel Cohen-Or. “An Image is Worth One Word: Personalizing Text-to-Image Generation using Textual Inversion”

---

> > > > ### Comment · Reviewer_xtbe · 2023-11-23
> > > >
> > > > Thanks for your detailed feedback, and I have raised the score to 5.
> > > >
> > > > I am sorry to forget the references and have presented them below.
> > > >
> > > > [1] Hu Ye, Jun Zhang, Sibo Liu, Xiao Han, Wei Yang. IP-Adapter: Text Compatible Image Prompt Adapter for Text-to-Image Diffusion Models.
> > > >
> > > > [2] Rinon Gal, et al. Encoder-based Domain Tuning for Fast Personalization of Text-to-Image Models. SIGGRAPH 2023.
> > > >
> > > > [3] Yuxiang Wei, et al. Elite: Encoding visual concepts into textual embeddings for customized text-to-image generation. ICCV 2023.

---

> > > > > ### Author Response · Authors · 2023-11-23
> > > > >
> > > > > Thank you for you comment and for raising our score.

---

### Official Review · Reviewer_AyFi · 2023-11-01

**Soundness:** 3 good
**Presentation:** 3 good
**Contribution:** 2 fair
**Rating:** 5
**Confidence:** 3

**Summary:**

The paper proposes an approach to update the text encoder of a text-to-image model to update factual knowledge within the text encoder (e.g., map "president of USA" from Donal Trump to Joe Biden). The update to the text encoder doesn't need additional training data and only requires very few parameters to be updated.

**Strengths:**

The paper is well written and well explained. The approach seems to be easy to implement, doesn't require additional training data, is done in closed form, and only changes a minimal amount of parameters.

The problem statement is also a realistic one in the sense that we don't want to retrain large models more often than absolutely necessary, so being able to update specific parts of them is useful.

**Weaknesses:**

I think a very simple baseline that is missing from the quantitative evaluation is to simply replace a concept/word with its intended new meaning, e.g., replace "President of the USA" with "Joe Biden" (or "phone" with "smart phone" etc). For most of the examples shown in the paper this would be a pretty straight forward approach to implement at large scale and wouldn't need any updates to the model parameters at all.

Also, it would be interesting to see how well the model handles more complicated real-world scenarios, e.g., what happens if someone uses another description for the president of the US (e.g., "American president", "head of the military", etc). Basically, it's not clear to me how well this approach translates to the complexities of the real world where it's not simply replacing one phrase with another phrase (which can already be achieved by the simple baseline I mentioned above). The generalization evaluation takes a step in that direction but I don't think it's general enough.

The same holds for specificity, for which I think a more general evaluation is necessary (again, sticking with the example above, what if the caption is simply "a photo of the president", would it show Joe Biden even though it doesn't specify that it should be the American president)?

**Questions:**

From a practitioner's point of view I wonder how well this scales to even more edits. Specifically, some edits might affect very similar parts of the text encoder, e.g., I might want to edit who is the president of multiple countries, would that still work?

Also, what are your thoughts on making the edits more context driven, e.g., apply a specific edit only if another condition is true (e.g., leaves of trees are green, unless the caption specifies it's autumn, in which case leaves should be brown)?

---

> ### Author Response · Authors · 2023-11-16
>
> We thank you for the thorough and thoughtful review. We address your concerns and questions below.
>
> 1. As for the baseline you discussed - this is indeed one of the approaches used in our experiments, referred to as the “Oracle”, which we discuss briefly in section 4.2.  While the oracle performs well, it does not achieve the goals of editing: the model itself still encodes outdated or incorrect associations and the extraction of the updated facts is left to the end user.
> The oracle is an unedited model, which generates a variation of the editing prompt that contains the target , in a similar manner to your suggestions. For example, if we wish to edit “The President of the United States” from “Donald Trump” to “Joe Biden”, an oracle prompt would be “Joe Biden as The President of the United States”. Our method, on the other hand,  is aimed for model providers to ensure the outdated or incorrect facts are no longer generated by the model, without requiring the end user to perform prompt engineering. The performance of the Oracle baseline is detailed in Table 1. We have added clarification on this approach in the paper by adding more details to section 4.2.
>
> 2. As for generalization - Figure 5 provides some examples of prompts that use synonyms and different phrasing of the editing prompt, such as prime minister/PM, cat/kitten, apple/granny smith and The tower of Pisa/Pisa tower. We have added additional discussion on this point in section 5.1.
>
> 3. As for specificity - In the case of “The President of the United States”, our test set includes negative prompts such as “A politician” and “A congressman” which are similar in nature to your suggestion. However, we find your question interesting and thus performed further tests with the prompts “A president” and “The President”, shown in Figure 19 in the Appendix. Before editing, using 25 different seeds, only 4 and 5 images (respectively, for each prompt) generated images of Donald Trump. After editing, the images only show minor changes and remain mostly the same as they were before edits, while seeds that portrayed Donald Trump before editing now portray a different person, representing a generic notion of “president” (not Joe Biden).
>
> 4. As for scaling our method to perform multiple edits - we performed experiments on editing up to 90 facts by applying ReFACT sequentially to the same instance of the model. Our results, discussed in section 5.4 and in appendix H, show that using ReFACT for sequential edits work almost as well as single edits in all three metrics.
>
> 5. As for context driven edits - this is indeed a very interesting use case. The context is encoded in the key k*, thus the edit is context dependent. This can be seen, for example, on the right in Figure 6, where we edit Ice cream to be strawberry ice cream. In this case images of ice are left unaffected, which shows that the context of “cream” leads to a specific edit.  In the case you suggested, we would have to better define what we expect the model to do in this case - if leaves are edited to be purple, do we still expect them to turn brown in the fall? Our experiment, added in Figure 17, shows that after editing “leaves” to “purple leaves” the prompt “an autumn leaf” indeed generates images of leaves that are purple, but a warmer autumn-like purple compared to the prompt “leaves”.
>
>
> We hope that we addressed your questions and concerns and are happy to continue the discussion.

---

> > ### Author Response · Authors · 2023-11-21
> >
> > Dear reviewer,
> >
> > Your acknowledgement of our response would be greatly appreciated, and we will gladly discuss your other concerns with you.

---

> > > ### Author Response · Authors · 2023-11-23
> > >
> > > Dear reviewer, with the discussion period ending today, we would greatly appreciate feedback on our response.

---

### Official Review · Reviewer_sFcd · 2023-11-01

**Soundness:** 3 good
**Presentation:** 3 good
**Contribution:** 3 good
**Rating:** 5
**Confidence:** 4

**Summary:**

The paper proposes to optimize a representation in text encoder to change the knowledge of text-to-image diffusion models.

**Strengths:**

1. It is an interesting task and show that it is possible to hack the text-to-image diffusion model and change the knowledge.

2. The generated results show that the method is able to replace the old concept with new ones.

**Weaknesses:**

1. The approach. The method carefully select the positive and negative prompts to update the representation layer to change the knowledge. The main problem is the selection. The objective may overfits the prompt and may not perform good at other cases. For instance, the second rows  in figure 1. show good results of the positive prompt in figure 3. It is unclear what would happen to other unshown prompts? For instance, can we generate "prince of wales is drinking coffee" and the sentence is not seen by the model? This result is necessary to the approach. Otherwise, we may just choose to add oracle desciption "William" to generate images rather than finetuning the model.

2. The evaluation. When authors perform evaluation on ROADS or TIME dataset, is the text-encoder updated every time once a new image is presented to the model? Or you copy the original model finetune the model for each concept? In addition, although authors show that the FID and CLIP is almost identical to the baseline model on the new datasets. It is necessary to include the FID and CLIP results on some benchmark text2image datasets to show that the model is able to generate other images after the tuning.


I'm happy to raise the score if above concerns are addressed.

**Questions:**

as above

---

> ### Author Response · Authors · 2023-11-16
>
> Thank you for your review and for taking the time to improve our work.
>
> 1. As for the positive and negative prompts - during editing, only three prompts are used: the editing prompt (e.g., “The Prince of Wales”), the source prompt (e.g., “Prince Charles”) and the target prompt (e.g., “Prince William”). To edit the model weights, we embed the given editing prompt into different prompt templates (e.g., “A photo of the President of Wales”) and get k*. To calculate v*, we use negative examples (prompts) from the MS-COCO dataset. During evaluation, we use a completely different set of negative prompts. Thus, a prompt like “The Prince of Wales drinking coffee” is completely unseen. Given this prompt, the model edited by our method generates correct images, containing Prince William. Further examples are shown in Figure 17, which we added to the paper.  Additionally, we have updated the paper and clarified  the terminology by now referring to the negative examples used for editing as “contrastive examples” and distinguishing them from the “negative prompts” which are a part of the dataset,  to avoid confusion.
>
> 2. The main results in the paper refer to the case of a single edit. In this case, a new “clean” pre-trained stable-diffusion instance is edited, and used to generate the images for evaluation. Using ReFACT to perform multiple edits, i.e. editing several associations on the same instance of the model is possible and discussed in section 5.4 and appendix H.
>
> 3. The FID and CLIP scores are indeed calculated on the MS-COCO dataset as is standard practice [1, 2, 3, 4] and as described in the last paragraph in section 4.3.
>
> We hope our response addressed your concerns and would be happy to provide any further clarifications. If our response is satisfactory, would you consider increasing your score?
>
> [1] Robin Rombach, Andreas Blattmann, Dominik Lorenz, Patrick Esser, and Bjorn Ommer. High-resolution image synthesis with latent diffusion models.
>
> [2] Chitwan Saharia, William Chan, Saurabh Saxena, Lala Li, Jay Whang, Emily Denton, Seyed Kamyar Seyed Ghasemipour, Raphael Gontijo-Lopes, Burcu Karagol Ayan, Tim Salimans, Jonathan Ho, David J. Fleet, and Mohammad Norouzi. Photorealistic text-to-image diffusion models with deep language understanding.
>
> [3] Aditya Ramesh, Prafulla Dhariwal, Alex Nichol, Casey Chu, and Mark Chen. Hierarchical text-conditional image generation with clip latents.
>
> [4] Yogesh Balaji, Seungjun Nah, Xun Huang, Arash Vahdat, Jiaming Song, Karsten Kreis, Miika Aittala, Timo Aila, Samuli Laine, Bryan Catanzaro, et al. eDiff-I: Text-to-image diffusion models with an ensemble of expert denoisers

---

> > ### Author Response · Authors · 2023-11-21
> >
> > Dear reviewer,
> >
> > Could you please acknowledge that you have read our response and let us know if you are satisfied with it or if you have any further concerns?

---

> > > ### Author Response · Authors · 2023-11-23
> > >
> > > Dear reviewer, as the discussion period concludes today, we would value your feedback on our response.

---

### Official Review · Reviewer_c2qA · 2023-11-09

**Soundness:** 3 good
**Presentation:** 3 good
**Contribution:** 2 fair
**Rating:** 5
**Confidence:** 4

**Summary:**

The text-to-image models often store factual information that can become outdated, limiting their usefulness. The authors proposed a new method -- ReFACT that can address this challenge by updating specific parts of the model without requiring direct user input or expensive re-training. The approach is evaluated on existing and newly created datasets and outperforms other methods in terms of adapting to related concepts while preserving unrelated ones.

**Strengths:**

The key strengths of the proposed method, ReFACT can be listed as follow

1.	Efficient Factual Updates: ReFACT efficiently updates factual information in text-to-image models without the need for extensive retraining, ensuring that the models stay up-to-date.

2.	Precision and Control: It allows for precise and controlled editing of facts, ensuring the accuracy of the desired factual updates.

3.	Superior Performance: ReFACT outperforms other editing methods, maintains image generation quality, and demonstrates strong generalization to related concepts, making it a highly effective tool for text-to-image model editing.

The paper is well-organized and the proposed method is easy to reproduce.

**Weaknesses:**

1.	The evaluation dataset is relatively small, and it would be beneficial to include a wider variety of prompts to evaluate ReFACT. For instance, additional prompts could involve questions about the current President of the United States or synonyms of the target prompts. This expanded evaluation would provide a more comprehensive assessment of ReFACT's performance and its ability to handle a diverse range of factual associations.
2.	The proposed method, ReFACT, appears to be straightforward in its approach to updating factual information in text-to-image models. However, the authors should clearly establish the differences between ReFACT and existing methods, such as "textual inversion." It is essential to provide a detailed comparison to highlight how ReFACT distinguishes itself.

**Questions:**

See the weakness section.

---

> ### Author Response · Authors · 2023-11-16
>
> We thank you for the thorough review and for taking the time to consider our work and for your valuable comments. We provide answers to your questions and address your concerns below.
>
> 1. As for the dataset size - indeed our evaluation data, consisting of our newly collected dataset ROAD and the prior dataset TIME, contains around 200 samples. Nonetheless, we argue that this test size is sufficient for showing improvement over the previous method and general effectiveness of our method. Furthermore, each test sample is composed of 11 different prompts: an editing prompt, five positive prompts and five negative prompts. Each prompt in the test sample is used for image generation with 25 different seeds, thus achieving stable results over each test sample. Overall, we generated more than 50k images for evaluation.
>
> 2. Variety of prompts: ROAD captures both roles and appearances, which add variety to TIME dataset’s implicit assumptions. In ROAD there are a variety of roles including politicians, musicians and tv characters, and appearances for objects such as recognized brands, plant varieties, and car models,  as discussed in section 4.1. We also find that our method is robust to synonyms and different phrasings of the editing prompt, such as prime minister/PM, cat/kitten, apple/granny smith and The tower of Pisa/Pisa tower. See Figure 5. We added an additional discussion on this point in section 5.1
>
> 3. As we discussed in section 2, personalization methods -- including Textual Inversion -- target a different task with different goals compared to editing. The main difference is that personalization methods add a special token (e.g., [v]) to distinguish the specific entity (“A [v] dog”) from the general class (“A dog”). Editing, however, should persistently alter the representation of the entity (e.g., “The President of the United States”) without requiring a specific token, and without preserving the original association (“Donald Trump”). Thus the two tasks are fundamentally different.
> Nonetheless, for comparison, we adapted Dreambooth, a personalization method which achieved superior performance on personalization datasets compared to Textual Inversion, to perform a variation of personalization that is related to editing (though not achieving the same goal).  As we discussed in Section 6 and Appendix I, we found that it leads to inferior performance compared to ReFACT, results in images that are less diverse and demonstrates catastrophic forgetting.
>
>
> We hope that we addressed your concerns, and are happy to continue a fruitful discussion.

---

> > ### Author Response · Authors · 2023-11-21
> >
> > Dear reviewer,
> >
> > We would appreciate your reply on our response. Do you have any other concerns?

---

> > > ### Author Response · Authors · 2023-11-23
> > >
> > > Dear reviewer, with the discussion period ending today, we would be grateful for your response.

---

### Author Response · Authors · 2023-11-16

We thank the reviewers for taking their valuable time to review our work and for striving to improve it. We are glad that the reviewers found the problem interesting (R2, R4), the method efficient (R1, R3, R4),  the paper well written (R1, R3), and that they recognized the contributions made by improving on previous methods (R1, R4). We addressed each of your concerns and questions individually. We have revised the paper to accommodate your comments and suggestions.

---

### Comment · Area_Chair_BDAe · 2023-11-20

Dear reviewers,

As the Author-Reviewer discussion period is going to end soon, please take a moment to review the response from the authors and discuss any further questions or concerns you may have.

Even if you have no concerns, it would be helpful if you could acknowledge that you have read the response and provide feedback on it.

Thanks,
AC

---

### Meta-Review · Area_Chair_BDAe · 2023-12-05

**Metareview:**

This paper proposes ReFACT for editing factual associations in text-to-image models. This main idea of the paper is to push the representation of the edit prompt to that of the target prompt. The proposed method updates the text encoder to reveal the changes of the factual representations by minimizing the the distance between the representations. The reviewers think that this paper considers an interesting and practical problem, but the technical contribution is limited. Specifically, the difference to Textual Inversion is not fully justified and the proposed method is confined to entities that could be described by texts. The AC agrees with the reviewers on the weaknesses, and therefore recommend a rejection for this paper.

**Justification For Why Not Higher Score:**

The reviewers have reached a consensus of a rejection (5, 5, 5, 5). In addition, given the weaknesses discussed above, the AC agrees with the reviewers that the overall contributions are limited, and some of the claims are not justified. Therefore, a reject is recommended.

**Justification For Why Not Lower Score:**

N/A

---

### Decision · Program_Chairs · 2024-01-16

Reject